# Nanoparticle enrichment mass-spectrometry proteomics identifies protein-altering variants for precise pQTL mapping

Karsten Suhre [1,2] ✉, Guhan Ram Venkataraman[3], Harendra Guturu[3], Anna Halama [1,2], Nisha Stephan[1], Gaurav Thareja [1], Hina Sarwath[4], Khatereh Motamedchaboki[3], Margaret K. R. Donovan[3], Asim Siddiqui[3], Serafim Batzoglou[3] & Frank Schmidt[4]

Proteogenomics studies generate hypotheses on protein function and provide genetic evidence for drug target prioritization. Most previous work has been conducted using affinity-based proteomics approaches. These technologies face challenges, such as uncertainty regarding target identity, non-specific binding, and handling of variants that affect epitope affinity binding. Mass spectrometry-based proteomics can overcome some of these challenges. Here we report a pQTL study using the Proteograph™ Product Suite workflow (*Seer, Inc.*) where we quantify over 18,000 unique peptides from nearly 3000 proteins in more than 320 blood samples from a multi-ethnic cohort in a bottom-up, peptide-centric, mass spectrometry-based proteomics approach. We identify 184 protein-altering variants in 137 genes that are significantly associated with their corresponding variant peptides, confirming target specificity of co-associated affinity binders, identifying putatively causal *cis*-encoded proteins and providing experimental evidence for their presence in blood, including proteins that may be inaccessible to affinity-based proteomics.

Large-scale studies of the plasma proteome using extensive biobanks have attracted increasing interest for their potential to inform drug development through insights gained from genome-wide disease association studies. Proteogenomic analyses identifying genetic variants associated with protein expression levels (protein quantitative trait loci, or pQTLs) can reveal proteins involved in key biological processes that affect complex traits and disease etiology[1].

The two main technologies employed to quantify protein levels in biological samples are affinity-based proteomics and mass spectrometry (MS)-based proteomics. Most large-scale proteomics studies to date have relied on affinity-based proteomics technologies, utilizing variations of Olink's antibody-based Proximity Extension Assays or Somalogic's aptamer-based SomaScan platform[2–7]. The UK Biobank Pharma Proteomics Project (UKB-PPP) quantified over 2900 proteins in blood plasma from 54,000 UKB participants using the Olink platform[8], and the deCODE study analyzed 4900 proteins in more than 35,000 individuals using the SomaScan technology[9], identifying tens of thousands of pQTLs.

Affinity-based proteomics approaches can deliver quantitative readouts for hundreds and even thousands of blood-circulating proteins in a high-throughput manner, but these approaches also possess certain limitations[1,10]. Notably, they are exposed to interference from genetic variants that can change the protein's epitope (structure) and modify the antibody or aptamer binding affinities, resulting in ambiguous or erroneous associations[1,10,11]. Additionally, establishing target specificity for affinity binders is challenging and must be determined on an individual basis under various physiological conditions. Although the literature considers a genetic association at the

[1]Bioinformatics Core, Weill Cornell Medicine-Qatar, Education City, 24144 Doha, Qatar. [2]Englander Institute for Precision Medicine, Weill Cornell Medicine, New York, NY 10021, USA. [3]Seer, Inc., Redwood City, Redwood City, CA 94065, USA. [4]Proteomics Core, Weill Cornell Medicine-Qatar, Education City, 24144 Doha, Qatar. ✉e-mail: kas2049@qatar-med.cornell.edu

gene locus that encodes the protein targeted by a given affinity-binder (*cis*-pQTL) as confirmatory evidence for target specificity, cross-reactivity with other proteins cannot be ruled out in such cases. Some protein classes may also be unsuitable for quantification by affinity binding (e.g., unfolded pro-peptides).

Epitope-modifying variants can result in false-positive associations between genetic variants and protein expression levels. Additionally, such variants often have a biological impact on protein function rather than on protein abundance. A recent study showed that approximately 50% of putative epitope-modifying variants colocalize with GWAS associations, suggesting that these variants modify protein properties and biological functions rather than protein abundance[5]. Consequently, genetic epitope effects caused by non-synonymous variation pose a significant challenge to the analysis and application of large-scale, affinity-based proteomics pQTL studies for drug development, because the effect of an epitope-modifying variant on the phenotype may not be through the protein expression level. Thus, therapeutic designs based on protein abundance might not yield the desired effect.

MS-based proteomics can help overcome some of the challenges of affinity-based proteomics by directly measuring variant peptides originating from protein-altering genetic variants. In a bottom-up MS-based proteomics approach, peptides (generated either by in silico digestion of a comprehensive protein database or curated experimentally) are matched against mass spectra collected by MS analysis of enzymatically digested proteins. Modern mass spectrometers, coupled with up-front liquid chromatography and ion mobility separation, enable the collection of hundreds of thousands of peptide fragmentation spectra at high mass-resolution in a data independent acquisition (DIA) mode[12]. MS-based proteomics methods can identify genetic epitope effects, as they provide peptide-level sequence readouts. Additionally, they may identify proteins that are not amenable to affinity binding and resolve potentially disease-relevant protein post-translational modifications.

However, bottom-up MS-based proteomics approaches also face some technological challenges related to peptide and protein identification and quantification[13–15]. In the context of pQTL studies, one such challenge is quantifying protein levels in the presence of genetic variation[16]. Current MS-based proteomics analyses typically ignore genetic variation, because incorporating all possible variants would significantly increase spectral library sizes and false-positive identifications. Consequently, standard proteomic libraries fail to detect variant peptides in homozygous alternate allele carriers and falsely suggest reduced protein levels in heterozygotes, leading to genotype-dependent (inaccurate) protein level measurements. This problem is exacerbated by instrumental and technical effects created by genotype-specific shifts in fragmentation, ionization, ion mobility, and liquid chromatography separation properties.

Here, we examine these technological challenges and propose potential solutions for effectively utilizing bottom-up MS-based proteomics in conjunction with available genetic variation information. We use the MS-based Proteograph™ Product Suite (Seer Inc.) workflow[17,18] to quantify protein and peptide intensities in blood samples from individuals in a multi-ethnic cohort. The Proteograph workflow uses five physicochemically distinct nanoparticles that each enrich for different proteins, thereby compressing the dynamic range of proteins analyzed downstream by DIA-MS[19]. Depending on protein abundance and biophysical properties, some peptides can be detected with two or more of the nanoparticles included in the Proteograph Assay. Detections by distinct nanoparticles can be considered technical replicates and offer additional internal validation of the data.

To account for genetic variability in peptide sequences, we implement a data analysis protocol (see Methods) that includes all single nucleotide protein-altering variants (PAVs) that are present in the study population at a minor allele frequency (MAF) higher than 10%. Given the size of our cohort, we expect at least 2–3 individuals to be homozygous for the minor allele at this level. We introduce these PAVs into the protein database (UniProt), translate the variants to amino-acid space, and perform in silico digestion. We then create three spectral libraries: one where we keep only the peptides that correspond to the reference alleles (termed the *reference* library), one where we include both reference and alternate peptides (termed the *PAV-inclusive* library), and one where we exclude all variant peptides, alternate and their respective reference peptides (termed the *PAV-exclusive* library). Note that the *reference* library corresponds to what is currently used in standard DIA-MS analyses. Using the three different libraries, we then quantify peptide and protein intensities using DIA-NN[15].

Next, we test the presence of the reference or alternate allele of PAVs for association with the presence or absence of the resulting variant peptide(s) in the proteome of the respective sample donor (detected with the *PAV-inclusive* library) using the Fisher's Exact test. Note that a PAV can give rise to multiple matching variant peptides, including peptides that differ by a single amino acid as well as more complex situations, e.g., when the PAV involves a trypsin cleavage site or a protein modification site. We use the term MS-PAV to refer to a PAV that associates significantly (after correcting for multiple tests) with its matching MS-detected variant peptide(s). We then ask whether the identified MS-PAVs also change the corresponding blood protein intensities. For this purpose, we test for association between the protein intensities (obtained using the *PAV-exclusive* library) with the copy number of the alternate allele of the respective MS-PAV as the dependent variable, as generally practiced in pQTL studies. We use the term MS-pQTL to refer to a PAV that associates with both the detection of the respective variant peptide (using the *PAV-inclusive* library) and the protein intensity (using the *PAV-exclusive* library). We show that pQTLs that have been identified by large affinity-based proteomics studies can be characterized by overlap between MS-PAVs and MS-pQTLs from MS-based proteomics studies.

Our study comprises the following steps: First, we identify MS-PAVs and MS-pQTLs using samples from a multiethnic clinical cohort. Then, we query the summary statistics of the two largest pQTL studies, which used the Olink and SomaScan platforms, respectively[3,9], to evaluate the power of this approach in identifying relevant pQTLs. Finally, we discuss new biological insights derived from this study by overlapping MS-PAVs and MS-pQTLs with GWAS associations with other phenotypes (Fig. 1).

## Results

### We identify 184 MS-PAVs by adding protein-altering variants to a bottom-up proteomics approach

Citrate plasma samples were obtained from 345 individuals who participated in the Qatar Metabolomics study of Diabetes (QMDiab)[20,21]. The previously unthawed samples (aliquot of 240 µL per sample) were processed using the Proteograph Product Suite (*Seer, Inc.*)[17,18] (see Methods). Briefly, samples were incubated with five proprietary physicochemically-distinct nanoparticles provided in the Proteograph Assay kit (*Seer, Inc.*) for protein corona formation. Nanoparticle-bound proteins were captured, digested using trypsin, and then analyzed using a dia-PASEF method[12] on a timsTOF Pro 2 mass spectrometer (*Bruker Daltonics*). All MS files were processed using the DIA-NN software (version 1.8.1) using library-free search with match-between-runs (MBR) enabled against the UniProt database (*reference*, accessed June 2022) and the derived *PAV-exclusive* and *PAV-inclusive* databases. The Proteograph workflow quantified 18,603 unique peptides from 2899 proteins detected in more than 20% of the samples (Fig. 2 and Supplementary Fig. 1).

In order to perform internal validation of the data prior to analysis, we evaluated the correlation between peptides that were

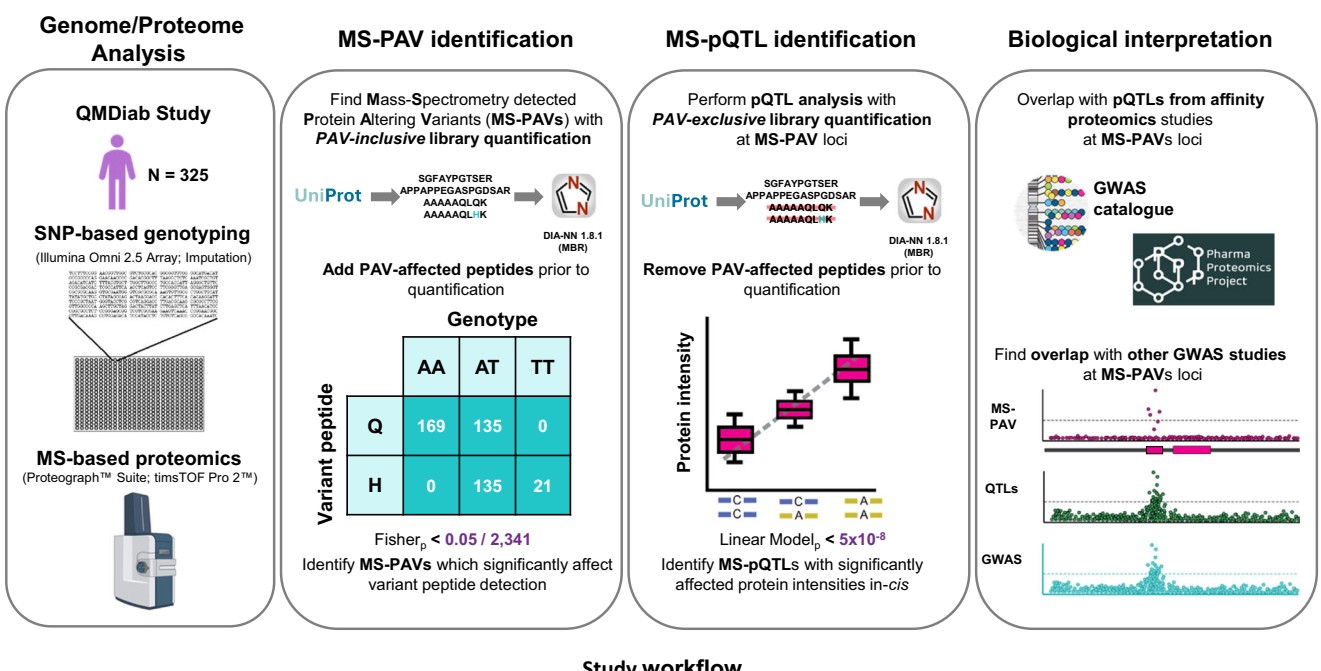

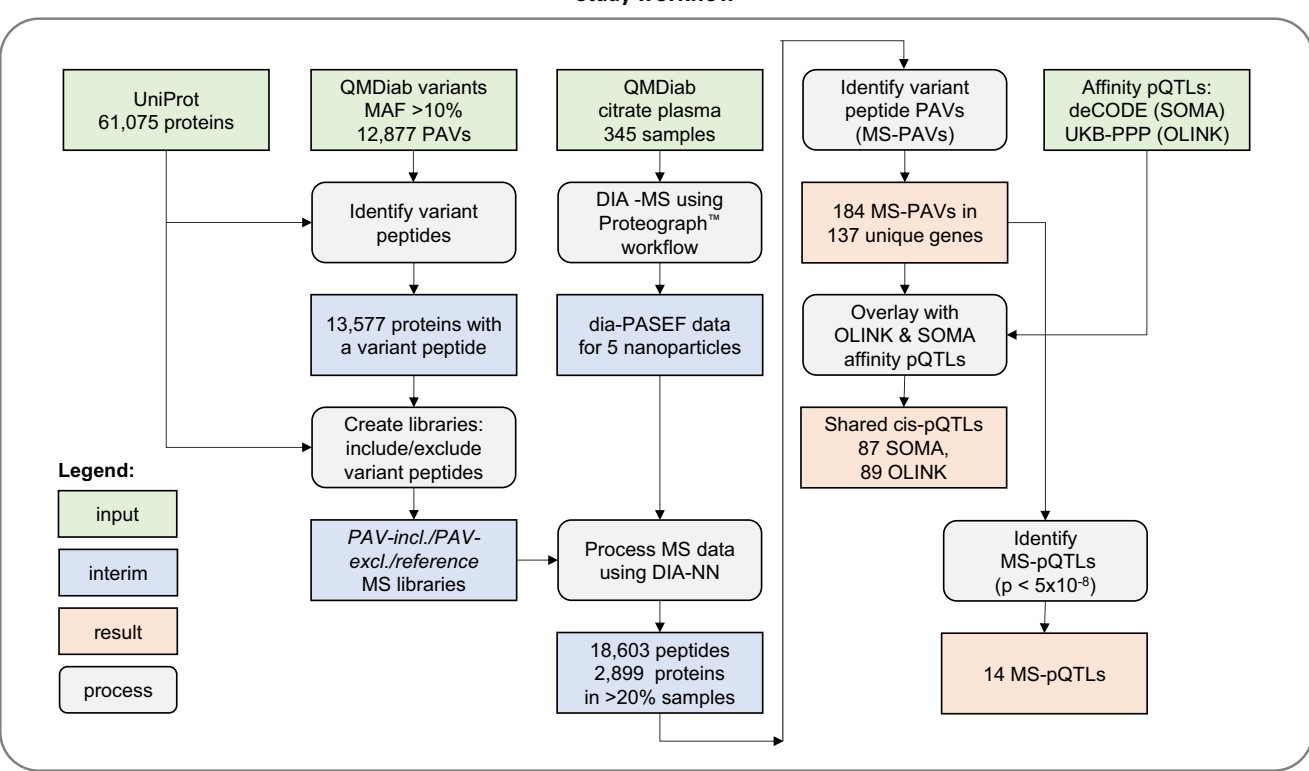

**Fig. 1 | Study design and workflow.** Procedure used to incorporate QMDiab variants into the UniProt.fasta file, create spectral libraries, and identify MS-PAVs and MS-pQTLs, plus an overview of the overall study design. Part of the Figure has been created with BioRender.com.

detected in more than one nanoparticle, limiting the analysis to 14,430 unique peptides (42,238 detections) detected using the *reference* library that had doubly-charged precursor ions and that were present in more than 20% of the samples. 3808 (26.4%) of these peptides were detected within a single nanoparticle fraction, while 4129 (28.6%) were detected in all five. 73.6% were detected more than once. The median Spearman correlation between a peptide measured in two or more nanoparticle fractions was rho = 0.67, and the Spearman correlation of a peptide measured in exactly two fractions was rho = 0.56 (Supplementary Fig. 2 and Supplementary Data 1).

Of the 345 analyzed samples, 325 were also genotyped on the Illumina Omni 2.5 platform and had imputed genotype data available[2,22]. Using the *PAV-inclusive* library, we identified 492 unique variant peptides that correspond to 2341 individual signals when accounting for detections related to different nanoparticles, precursor charges, and missed cleavages (Supplementary Data 2). These variant peptides mapped to 317 distinct genetic variants in 251 genes. To filter to a set of reliably detected variant peptides and avoid false positives, we asked whether each peptide's detection matched the individual blood donor's genotype. A total of 1000 of the 2341 variant peptide

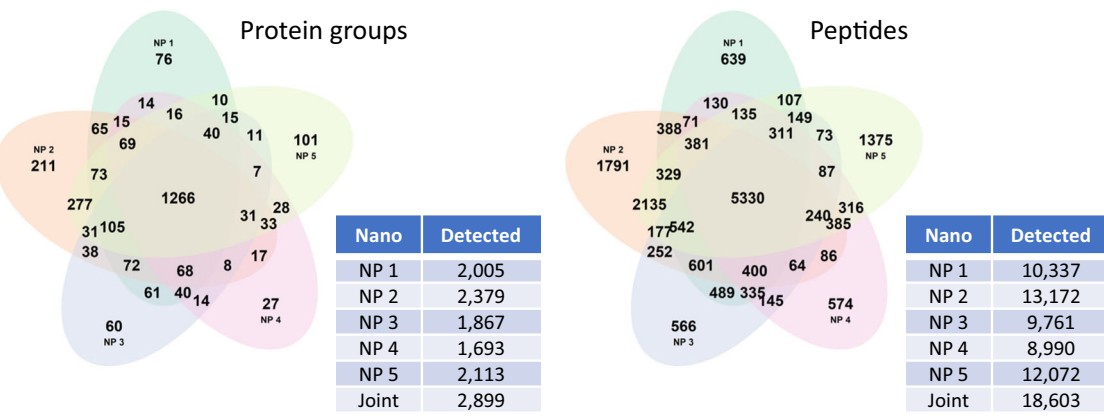

**Fig. 2 | Proteins and peptides detected in > 20% of the samples by the Proteograph™ workflow.** Data is for protein and peptide identification using DIA-NN with the *reference* library using the match-between-runs (MBR) option (see Supplementary Fig. 1 for the dependence between number of detections and % missingness).

detections were significantly associated with the genotype of the coding variant in a Fisher's Exact test at a Bonferroni level of significance of $p < 2.1 \times 10^{-5}$ (0.05/2341). Note that most of the non-significant variant peptides had a low detection frequency and did not provide sufficient statistical power to reach the required significance level; 512 had seven or less detections, while only peptides with eight or more detections reached Bonferroni significance. The 1000 significant associations corresponded to 306 unique variant peptides that were generated by 184 unique MS-PAVs. These MS-PAVs were located in 137 different genes (Supplementary Data 3).

## Robust pQTLs can be identified by excluding variant peptides from the spectral library

In MS-based proteomics, protein intensities are generally inferred from the intensities of one or multiple peptides that are derived from that protein. When peptides map ambiguously to multiple proteins, inference algorithms group them together into so-called protein groups and quantify them jointly. When a peptide that is used for protein quantification contains an amino acid-changing genetic variant (an MS-PAV), the resulting protein level will reflect the genotype of the sample donor and result in a spurious pQTL; this phenomenon can be considered the MS equivalent of an epitope effect in affinity-based proteomics. We counter this issue by excluding variant peptides from the protein quantification process. This exclusion will not necessarily reduce pQTL sensitivity for MS-PAVs that directly alter the corresponding blood protein level or for those MS-PAVs in linkage disequilibrium with regulatory variants controlling the protein's gene expression (MS-pQTL). In these cases, all peptides derived from the protein that do not overlap with the position of the MS-PAV are expected to vary in the same way as the genotype and to equally reflect the protein level.

Figure 3 demonstrates the impact of including or excluding PAVs in the process of protein quantification in the example of the Factor V (F5) protein. When using the *PAV-exclusive* library, no pQTL is observed at the protein level, which is consistent with the absence of pQTLs on the non-PAV containing peptides. When using the *PAV-inclusive* library, the two F5 MS-PAV isoforms that correspond to a K > R substitution (rs4524) are identified as pQTLs. This is also expected, as we are considering the expression level of each isoform separately in this case. No pQTLs are identified for any of the other peptides. However, when using the *reference* library, a pQTL is found, which is incorrect because the overall level of F5 protein does not vary with genotype. This is an example of the MS equivalent of an epitope effect in affinity-based proteomics, but here, in contrast to affinity-based proteomics, MS-based proteomics can easily capture and interrogate these effects. We

performed a pQTL analysis at the 184 MS-PAVs and calculated their associations with their corresponding protein intensities derived using the *PAV-exclusive* library (Supplementary Data 3). Fourteen MS-PAVs reached a genome-wide significance level of *p*-value $< 5 \times 10^{-8}$ and are considered as MS-pQTLs (Table 1). Four of these were not identified by either Olink or SomaScan in the largest pQTL studies conducted with each; seven were cross-verified by SomaScan; and eight were cross-verified by Olink.

We then compared the association statistics with those obtained using the *reference* library. Robust MS-pQTLs are on the diagonal of the scatterplot presented in segment 1 of Fig. 4. Instances in which non-significant results from the *PAV-exclusive* library overlap with significant results in the *reference* library indicate situations where the current standard approach fails (segment 2 in Fig. 4). If these variants overlap with *cis*-pQTLs from affinity-based proteomics studies, they reveal potential epitope effects. MS-PAVs that do not reach significance using either library are MS-PAVs that do not lead to detectable changes in protein expression in our cohort (segment 3 in Fig. 4); however, our cohort size is relatively small, and larger sample sizes are needed to reach the statistical power required to determine whether these MS-PAVs are also MS-pQTLs. These observations suggest that: a) MS-PAVs can be detected at the peptide level by using the *PAV-inclusive* library and conducting a Fisher's Exact test performed on the presence/absence of the coding variant versus MS detection/non-detection of the corresponding variant peptide; and b) using a *PAV-exclusive* library to generate pQTLs is essential to prevent false-positive pQTL identifications (i.e., avoid the MS equivalent of epitope effects in affinity-based proteomics).

## Overlap of pQTLs from affinity-based proteomics platforms

We then investigated which of the 184 MS-PAVs had been identified in previous pQTL studies and which were new (Supplementary Data 3). We identified overlapping pQTLs using summary statistics from the deCODE (SomaScan)[9] and UKB-PPP (Olink)[3] studies (Table 2 and Supplementary Fig. 4). To identify overlapping eQTLs and pQTLs that were not covered by these two large studies, we further annotated the 184 MS-PAVs using Phenoscanner[23] ($r^2 = 0.8$ using linkage data from the EUR population, accessed 21 Jan 2023) and omicsciences.org[6].

Two thirds of the MS-PAVs (124) overlapped with *cis*-pQTLs previously identified by affinity-based proteomics pQTL studies, thus confirming the target specificity of the affinity binders. An additional 42 MS-PAVs overlapped with *trans*-pQTLs (and not *cis*-pQTLs) in affinity-based proteomics studies, thus identifying the putatively causal *cis*-encoded protein and providing experimental evidence for its

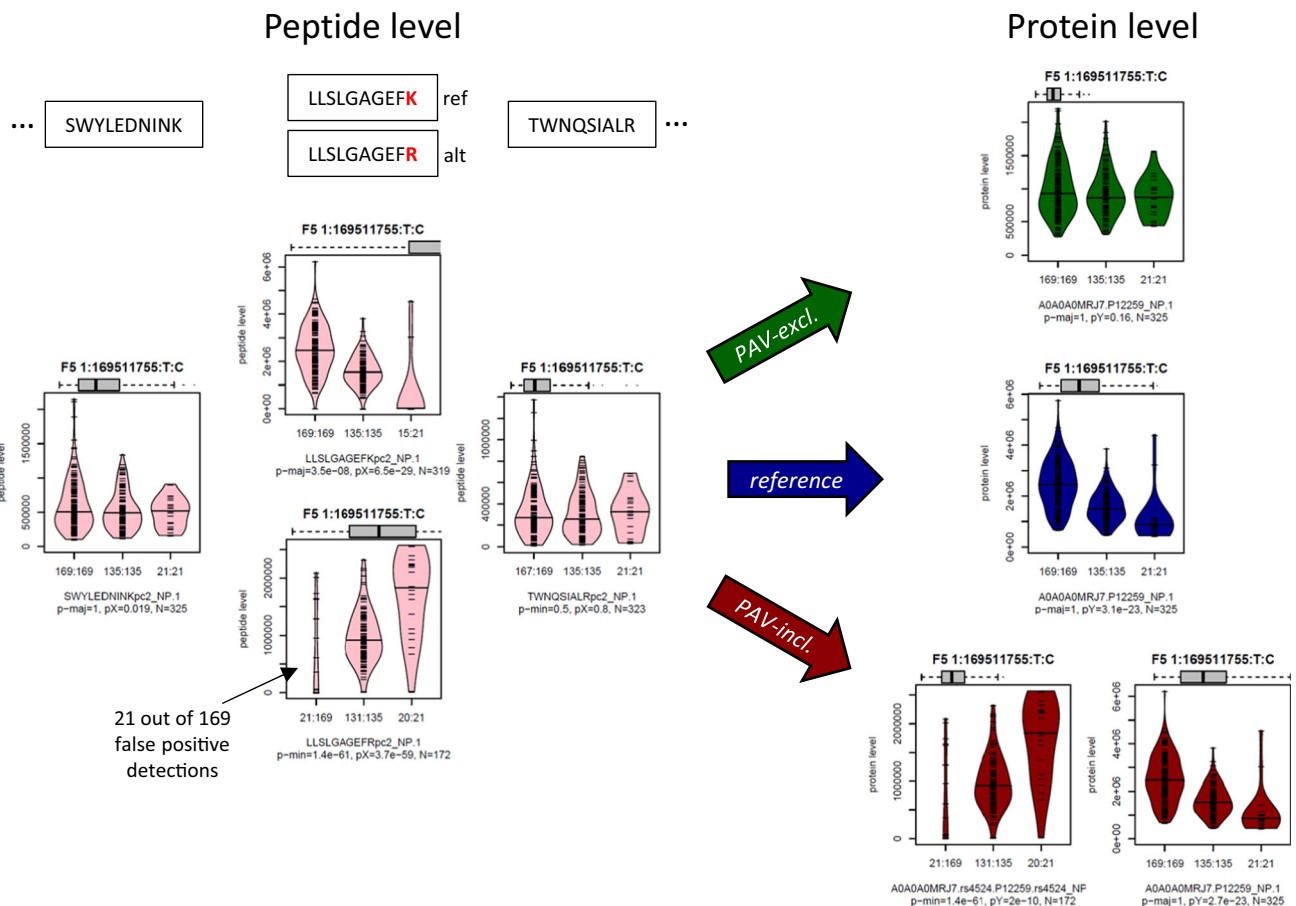

**Fig. 3 | Boxplots by genotype rs4524 for selected Factor V (F5) protein and peptide intensities.** This figure shows the effect of using the different libraries at the example of the Factor V (F5) protein. Similar plots are provided in Supplementary Fig. 3 and as Source Data with this paper for all 184 MS-PAVs; The boxes are color-coded as following: using the *PAV-exclusive* library (green), using the *reference* library (blue), and using the *PAV-inclusive* library (red). Protein intensities are in dark colors, and peptide intensities are in light colors. The grey horizontal boxplots on top of the plots represent the range of the data shown in that plot compared to the 5–95% range of the entire data for that protein. Units on the *y*-axis are engine-normalized intensities as provided by DIA-NN. The *x*-axis labels indicate the number of detected peptides/proteins followed by a colon and the number of samples with the given genotype (order: reference/major allele homozygote, heterozygote, alternate/minor allele homozygote). The first line of the subtitle identifies the protein (Uniprot ID and rsID, when applicable) or the peptide sequence followed by the nanoparticle used in that analysis. The second line shows the number of data points included in generating the plot (*N*). Significance levels (*p*-values) for the following hypothesis tests are given: (1) Fisher's Exact test on detected/non-detected versus presence/absence of the major (p-maj) or minor (p-min) allele, where the stronger of the two associations is shown (indicating MS-PAV detection significance), and (2) a linear regression of peptide intensity versus genotype (coded 0-1-2) with missing values set to zero (pX), and for proteins a linear model including relevant covariates using inverse-normal scaled protein intensities (excluding missing values) against genotype (pY; indicating pQTL significance). Protein name, chromosome, chromosome position (GRCh37), and major and minor alleles are indicated in boldface on top of the boxplots.

presence in blood. The remaining 18 MS-PAVs were novel and included proteins that may be inaccessible to affinity-based proteomics, such as a variant in the incretin pro-peptide (Gastric Inhibitory Polypeptide, GIP) which plays a central role in type 2 diabetes and cardiovascular disease pathologies.

Out of the 184 MS-PAV variants, 145 shared a pQTL with at least one target of the SomaScan platform. In 87 cases, these were *cis*-pQTLs. For Olink, we identified 148 overlapping pQTLs, 89 of which were located in-*cis*. 59 MS-PAVs were *cis*-pQTLs in both the Olink and SomaSan study, and 117 were *cis*-pQTLs on at least one platform. Of 67 MS-PAVs with no matching *cis*-pQTL, neither in the deCODE nor the UKB PPP study, seven matched a *cis*-pQTL in another study as identified by Phenoscanner or omicsciences.org. This leaves a total of 60 MS-PAVs detected using the Proteograph workflow that were not previously identified as cis-pQTLs in any large-scale affinity-based pQTL studies. These 60 novel MS-PAVs were located within 52 unique genes.

Taken together, the observations highlight the complementarity between the affinity- and MS-based proteomics approaches.

## Specific findings using the proteograph workflow

We annotated the 184 MS-PAVs with overlapping expression QTLs (eQTLs), metabolomics QTLs (mQTLs) and GWAS associations (Supplementary Data 3). Out of the 184 MS-PAVs, 121 match an eQTL reported in Phenoscanner, suggesting that these variants not only influence the peptide sequence but also alter the corresponding gene expression levels. 90 MS-PAVs overlap a GWAS hit (not counting metabolite/protein levels and body height), including 27 of the variants that do not have a *cis*-pQTL on the SomaScan and Olink platforms (Table 3).

For example, variant rs2291725 corresponds to an S > G amino acid exchange in the Gastric Inhibitory Polypeptide (GIP). This variant is in the *GIP* gene, which codes for an incretin hormone and stimulates insulin secretion. The amino acid exchange occurs in a peptide consisting of ten amino acids (ALELA[S/G]QANR) on the incretin pro-peptide (aa98-107). This part of the protein is not part of the processed incretin hormone. The variant peptide corresponding to the alternate allele is detected in 130 out of 223 carriers of the alternate allele, together with five false-positive detections in

102 reference allele homozygotes ($p = 1.2 \times 10^{-22}$, Fisher test). This variant is associated with several body fat traits, coronary artery disease, and diabetes in the GWAS catalog, consistent with *GIP's* function and suggesting a causal role for this variant in these clinical phenotypes. This *GIP* variant has not been reported by any affinity-based proteomics GWAS before, possibly because GIP is too small or transiently folded and may not be detected by affinity binders. GIP agonism has recently gained renewed attention as a satiety-suppressing drug (similar to GLP-1 inhibitors, but with possibly less severe side effects such as nausea)[24]. Hence, this variant may serve as a potential genetic instrument to further investigate the potential effects of GIP inhibition.

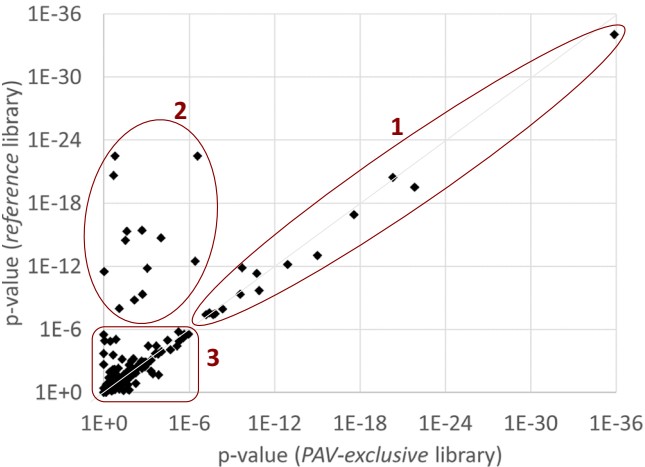

**Fig. 4 | Scatterplot of the protein-level associations (*p*-values) for the 184 MS-PAVs using the *reference* and the *PAV-exclusive* libraries.** Three regimens are labeled: (1) variants that remain associated with protein levels after removal of the variant peptides from the library (MS-pQTLs), (2) variants where the association signal with the protein levels disappears after removal of the variant peptides (the MS equivalent of an epitope effect), and (3) variants that do not associate with protein levels in either case (MS-PAVs that may become significant in more highly powered studies). Plot data is in Supplementary Data 3. *P*-values (unadjusted) are from linear model as described in the methods section.

Another key protein relevant to cardiovascular disease is APOB. Previous GWAS studies associated genetic variation in *APOB* with many relevant lipid-related traits, with lead associations with LDL-cholesterol (LDL-C) and Apolipoprotein B (ApoB) levels measured by clinical biochemistry[25]. We found that the variant rs1367117 (chr2:21263900) associated with the alternate and also the reference allele of the ApoB variant peptide TSQC[T/I]LK (*p*-value = $1.0 \times 10^{-68}$ and $3.5 \times 10^{-16}$, respectively; Fisher's Exact test), but we did not detect a significant association signal at the protein level (*p*-value > 0.02). To analyze this association in its genetic context, we computed the associations between the detection of peptide TSQCILK and all variants in the vicinity (+/−250 kb) of this MS-PAV, retrieved GWAS data for the associations with clinical biochemistry measures of LDL-C and ApoB in the UK Biobank, and generated regional association plots (see Methods, Fig. 5). The regional association plots show that rs1367117 has the strongest LDL association, but they also indicate the presence of at least one additional equally strong association between a variant in the promoter region of *APOB* with both LDL and ApoB levels. This observation suggests the presence of two distinct signals, one likely acting via a structural change in the ApoB protein itself, and a second that may be attributed to changes in ApoB protein levels. Interestingly, we previously found that these two signals also lead to distinct phenotypes in lipoprotein composition[26]. Our study is the first study that directly identifies this putatively causal genetic variant of high LDL-C levels at the peptide level using MS-based proteomics at a population scale and shows how MS-PAVs can be used to dissect complex genetic association signals.

**Proteome-wide association study of non-genetic determinants**
To explore whether using our method also improves associations with non-genetic determinants, we conducted an association study of proteins quantified using the reference and the *PAV-exclusive* libraries with age, sex, diabetes state, and BMI. We included 3,657 protein group/nanoparticle combinations in the analysis that were detected in > 80% of the shared samples. We found that the majority of proteins (3183, 87.0%) correlated strongly between the two methods (Spearman rho > 0.8) while only few (91, 2.5%) changed substantially when using the different libraries (Spearman rho < 0.5) (Supplementary Fig. 5). We then computed linear models including age, sex, diabetes state, BMI, and the first three genotype principal

## Table 1 | MS-pQTLs

| Gene | UniProtID | rsid | SNP | MAF | *p*-value | beta | *cis*-pQTL |
|---|---|---|---|---|---|---|---|
| MST1 | G3XAK1 | rs3197999 | 3:49721532:G:A | 23.4% | 2.6E-18 | −0.716 | Both |
| ITIH1 | P19827 | rs1042779 | 3:52821011:A:G | 37.7% | 1.2E-11 | 0.545 | Both |
| KNG1 | P01042 | rs2304456 | 3:186445052:T:G | 16.0% | 4.8E-21 | 0.911 | SomaScan |
| HLA-C | A2AEA2 | rs707908 | 6:31238053:G:C | 21.8% | 2.0E-10 | 0.630 | None |
| CFB | B4E1Z4 | rs12614 | 6:31914179:C:T | 18.5% | 1.2E-13 | −0.590 | Both |
| PON1 | P27169 | rs662[a] | 7:94937446:T:C | 37.4% | 4.6E-09 | −0.424 | Olink |
| PON1 | P27169 | rs854560[a] | 7:94946084:A:T | 27.4% | 2.7E-10 | 0.499 | Olink |
| PON2 | A0A0J9YXF2 | rs12026 | 7:95041016:G:C | 30.6% | 1.3E-36 | −0.998 | Olink |
| FGL1 | Q08830 | rs3739406 | 8:17739538:T:C | 49.2% | 1.8E-11 | 0.512 | Both |
| GALC | G3V255 | rs34362748 | 14:88442712:C:T | 11.5% | 1.0E-15 | 0.887 | None |
| SERPINA10 | G3V2W1 | rs2232700 | 14:94756450:T:A | 30.8% | 1.5E-22 | 0.741 | SomaScan |
| SERPINA1 | P01009 | rs709932 | 14:94849201:C:T | 23.1% | 3.8E-08 | 0.431 | Both |
| DSC3 | Q14574 | rs276938[b] | 18:28610988:C:T | 41.4% | 2.1E-08 | −0.422 | None |
| DSC3 | Q14574 | rs276937[b] | 18:28611061:A:T | 41.2% | 1.5E-08 | −0.429 | None |

Associations of MS-PAVs that are significantly ($p < 5 \times 10^{-8}$) associated with protein levels derived using the *PAV-exclusive* library; detection of a *cis*-pQTL on the SomaScan and/or Olink platform is indicated.
[a]correlation between rs662 and rs854560 is $r^2 = 0.20$.
[b]correlation between rs276938 and rs276937 is $r^2 = 0.99$.

**Table 2 | Overlap of PAVs with *cis*- and *trans*-pQTLs on the SomaScan and Olink platform**

| | deCODE (SomaScan) | UKB-PPP (Olink) | In either studies | In both studies |
|---|---|---|---|---|
| # of assayed proteins | 4660 | 2908 | - | - |
| # samples in study | 35,446 | 51,637 | - | - |
| Overlapping PAV variants | 182[a] | 184 | 184 | 182 |
| PAV proteins assayed by platform | 127 | 100 | 143 | 84 |
| PAV proteins not assayed by platform | 55 | 84 | 41 | 98 |
| Overlapping significant association with PAV protein (*cis*-pQTL)[b] | 87 | 89 | 117 (124[d]) | 59 |
| Overlapping association with another protein, but not the PAV protein (*trans*-pQTL)[c] | 58 | 59 | 87 | 30 |
| Overlapping significant association with any protein (*trans*- or *cis*-QTL) | 145 | 148 | 166 | 143 |
| No overlapping pQTL | 37 | 36 | 41 | 18 |
| No overlapping *cis*-pQTL | 95 | 95 | 125 | 67 (60[d]) |
| No overlapping *cis*-pQTL found, although the PAV protein was assayed | 40 | 11 | 6 | 45 |
| No overlapping *cis*-pQTL found, because the PAV protein was not assayed | 55 | 84 | 119 | 22 |

Data and further details supporting these numbers are in Supplementary Data 3.

[a]Variants 3:52853480:T:A and 6:32552029:A:T were missing from the deCODE summary statistics.

[b]*cis*-pQTL in Supplementary Data 3; *p*-value < 0.05/182 for SomaScan and < 0.05/184 for Olink.

[c]*trans*-pQTL in Supplementary Data 3; *p*-value < 0.05/182/4660 for SomaScan and < 0.05/184/2908 for Olink.

[d]including 7 *cis*-pQTLs identified by PhenoScanner and omicsciences.org.

**Table 3 | MS-PAVs that overlap with disease-relevant GWAS hits**

| Gene | UniProtID | rsID | Fisher P | Peptide | GWAS trait |
|---|---|---|---|---|---|
| WDR1 | O75083 | rs13441 | 4.8E-37 | FTIGDHSR | Atrial fibrillation and flutter |
| PIP4K2A | H7BXS3 | rs2230469 | 2.2E-48 | IYIDDNSK | Body fat composition |
| CHID1 | Q9BWS9 | rs6682 | 5.9E-59 | MVWDSQASEHFFEYK | Body mass index |
| SCFD1 | A0A7I2V362 | rs229150 | 1.8E-28 | FGQDIISPLLSVK | Amyotrophic lateral sclerosis |
| SPTB | P11277 | rs229587 | 9.6E-14 | ETWLNENQR | Red blood cell phenotypes |
| LOXL1 | H3BUV8 | rs1048661[a] | 1.0E-70 | EVAVGDSTGMALAR | Exfoliation glaucoma |
| LOXL1 | H3BUV8 | rs3825942[a] | 5.5E-57 | HGDSASSVSASAFASTYR | Coronary artery disease, Exfoliation glaucoma |
| GIP | P09681 | rs2291725 | 1.2E-22 | ALELAGQANR | Coronary artery disease, Type II diabetes |

Selected MS-PAVs that have not been reported as *cis*-pQTLs in previous pQTL studies and that overlap with a clinically relevant GWAS catalog entry (edited for brevity, see Supplementary Data 3 for details).

[a]correlation between rs1048661 and rs3825942 is $r^2 = 0.12$.

components as determinants and inverse-normalized protein levels as outcome. We found many previously reported associations, such as associations between leptin and sex and CRP and BMI, and also new associations that are biologically plausible, such as LIPL and LIPE with diabetes status (Supplementary Fig. 6). However, we did not find evidence that using the *PAV-exclusive* library strengthens the associations between proteins and these non-genetic determinants. The fact that we did not see an improvement with these non-genetic determinants is likely because these variations may affect individual peptides/protein quantifications in a substantial manner so as to affect the number and spread of pQTLs found, but their effects may be too granular to affect phenotype. We provide these associations in Supplementary Data 4.

## Discussion

To our knowledge, this is the first cohort proteogenomic analysis in which genetic variation has been systematically investigated at the peptide level using a mass spectrometry-based, bottom-up proteomics approach. We show that MS-based proteomics has the potential to access genetic variation in proteins at the peptide level and to complement affinity-based proteomics pQTL studies by: a) providing additional information on protein identity and potential epitope effects, b) assessing proteins that are not accessible to affinity binding, and c) incentivizing future applications that elucidate post-translational modifications and protein group resolution.

Our study also has limitations. First, by excluding peptides from the *PAV-exclusive* library, some of the MS spectra remain unaccounted for and can yield false-positive matches to other peptides in the library. Future approaches could remove variant peptides only during the protein quantification step. This would also reduce the effort needed for identifying peptides with multiple libraries. In addition, new quantification algorithms could use data from all nanoparticles in parallel.

Another limitation is the choice of the MAF cutoff. Rarer variants are not detected, because including lower-frequency variants could lead to a significant increase in false-positive detections. The inclusion of rare variants may also lead to multiple amino acid changes within the same peptide simultaneously, which we do not presently account for. Using sample-specific libraries that account for individual genetic variants can mitigate this problem in the future. If these libraries additionally used phased genotype data, potential issues when two variants are located on the same peptide could be addressed.

We observe some cases where the detected variant peptide does not match the PAV, and a few isolated cases where the alternate and reference alleles are both detected in all samples, such as an E > D substitution in Complement Factor H (CFH peptide SPP[E/D]ISHGV-VAHMSDSYQYGEEVTYK). These false positive identifications (Supplementary Fig. 7) can be attributed to uncertainties or shortcomings in the algorithms that match the MS2 spectra of alternate and reference peptides that occur in the same DIA-MS window and share many

common fragments. Due to the very low error rate in today's genotyping platforms, genetic variants can comparatively be considered a ground truth to calibrate peptide detection algorithms. These false positive identifications are an inherent challenge for all bottom-up MS data analysis softwares and are not specific to our workflow. We suggest that these algorithms may be improved in the future by using combined genetic and proteomic data from studies like ours as a benchmark.

The level of MS-PAV detection (137 out of 2899 quantified proteins) is in line with expectations for several reasons. The entire protein library contains 61,075 protein entries, of which only 13,577 (22.2%) had at least one peptide with a coding variant with MAF > 10%. We detect variant peptides in 492 (17.0%) of the 2899 proteins. The difference can be explained by peptides that do not ionize well enough to reach the detector or that do not contain suitable cleavage sites with Trypsin/Lys-C enzymatic digestion to form peptides that can be detected with MS. The fact that 137 of the 492 detected peptides reached the required Bonferroni level of significance in the Fisher exact test can be explained by statistical power, as most of the non-significant variant peptides had less than 10 detections.

Our study highlights both the complementarity and the complexity of affinity- and MS-based proteomics in pQTL discovery and suggests a new approach to analyzing MS-based proteomics data in the presence of genetic variation. We propose to use naturally occurring genetic variation for the development of future and more powerful MS-based proteomics data analysis tools. Deployed at scale, this approach can provide valuable new insights for drug target prioritization and repurposing.

## Methods

### Ethics

This study was approved by the institutional research boards of Weill Cornell Medicine – Qatar under protocol #2011-0012 and of Hamad Medical Corporation under protocol #11131/11 and complies with all relevant ethical regulations. For forthgoing work with the study a non-human subjects research determination was obtained. The study design and conduct complied with all relevant regulations regarding the use of human study participants (in this case, human material and data) and was conducted in accordance to the criteria set by the Declaration of Helsinki.

### The QMDiab study

The Qatar Metabolomics study of Diabetes (QMDiab) was conducted in 2012 at the dermatology department of Hamad Medical Corporation, the major public hospital in Doha, Qatar, with the primary aim to study metabolic differences in individuals with and without diabetes in

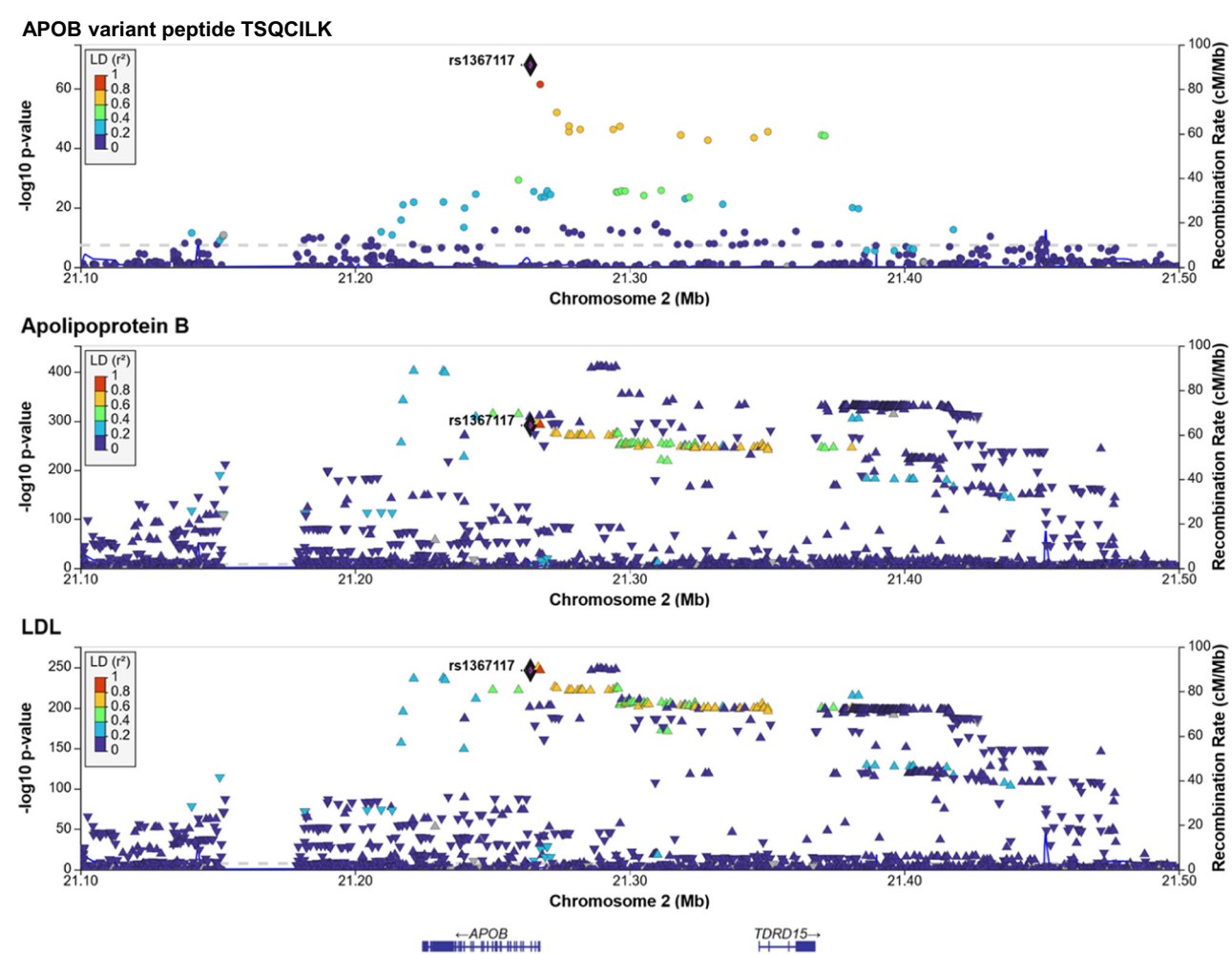

**Fig. 5 | Regional association plots for the APOB region.** Association of the detection of the alternate variant peptide TSQCILK of APOB (pc2, nanoparticle 1) with the presence/absence of the matching genetic variants at the APOB locus (top), GWAS associations of Apolipoprotein B (middle) and LDL-cholesterol (bottom) measured by clinical biochemistry methods in blood samples from 343,621 participants of the UK Biobank study. The highlighted variant rs1367117 (chr2:21263900) is the MS-PAV in TSQC[T/I]LK. Linkage (LD) between variants is indicated color, gene positions are below. *P*-values (unadjusted) are from linear models generated by the respective studies.

adult female and male participants of Arab and Asian ethnicities[20,21]. A total of 391 study participants were enrolled, sample material from 345 of them was assayed on the *Seer Inc.* proteomics platform. Cases and controls, males and females, Indian, Arab and Filipino ethnicities were represented in equal proportions covering an age range from 18 to 80. Multiple aliquots of blood, urine and saliva samples were collected and stored at −80 °C without further freeze-thaw cycles.

## Genotyping
DNA from QMDiab samples was extracted and genotyped using the Illumina Omni 2.5 array (version 8) and imputed using the SHAPEIT software with 1000 Genomes (phase3) haplotypes. PAVs were identified in the imputed variant dataset using the Ensembl Variant Effect Predictor (VEP)[27] and filtering to MAF > 10%. Genotyping data was available for 325 of the 345 analyzed samples.

## Gene model alignment
The gene model was constructed using the June 2022 version of the UniProt.fasta file and the UniProt genome annotation tracks UP000005640_9606 [https://ftp.uniprot.org/pub/databases/uniprot/current_release/knowledgebase/genome_annotation_tracks/UP000005640_9606_beds/UP000005640_9606.proteome.bed] (UniProtgenome, accessed June 2022). To create the gene model, we aligned the.bed with the.fasta they provide. We kept UniProt IDs that unambiguously mapped to one sequence. For those that mapped to multiple sequences, we preferentially selected those sequences that aligned perfectly when translating the.bed coordinates using the GRCh37.fasta file. For those that did not map to the.fasta, we preferentially selected sequences that started with Methionine and were in-frame. We removed ambiguity in UniProt IDs that had sequences in multiple chromosomes by picking the canonical one if available, and then alphanumerically if not. For those UniProt IDs that had multiple canonical sequences within the same chromosome, we picked the first sequence within the gene model.

## Library construction
The gene model file from UniProt was used to generate reference sequences for every UniProt ID. Common (MAF > 10%) protein-altering variants were identified using the Ensembl Variant Effect Predictor (VEP)[27] and injected into the corresponding reference protein sequences. We digested the reference and alternate sequences in a manner akin to DIA-NN in silico (i.e., on tryptic [K/R] amino acids; with/without one missed cleavage; and peptide length between 7 and 30 AAs) to generate constituent peptides per UniProt sequence. We then compared the digests from the corresponding reference and alternate gene sequences. If peptides were of equal length, shared their initial position within the full gene, and differed in sequence, then the peptides were declared a reference-to-alternate match; otherwise, they were annotated as complex (indicated in Supplementary Data 3). We repeated this process with and without 1 missed cleavage. All other mismatched injected variant sequences (which were a result of the introduction or deletion of a K/R), were discarded. For each variant, a protein entry with the corresponding amino acid exchange was also added to the *PAV-inclusive* library as an isoform using the protein identifier (UniProt ID) followed by the variant identifier (rsID). Similarly, the corresponding reference sequences were discarded from the *PAV-exclusive* library. Genetic variants were considered independent, and only one variant per protein was considered at a time to avoid combinatorial growth of the library.

## Proteomic analysis
240 μL of previously un-thawed citrate plasma were loaded onto the SP100 Automation Instrument for sample preparation with Proteograph Assay Kits and the Proteograph workflow[17,18] (Seer, Inc.) to generate purified peptides for downstream Liquid Chromatography coupled with Mass Spectrometry (LC-MS) analysis. Each plasma sample was incubated with five proprietary, physicochemically-distinct nanoparticles for protein corona formation. Samples were automatically plated, including process controls, digestion control, and MPE peptide clean-up control. A one-hour incubation resulted in a reproducible protein corona around each nanoparticle surface. After incubation, nanoparticle-bound proteins were captured using magnetic isolation. A series of gentle washes removed non-specific and weakly-bound proteins. The paramagnetic property of the nanoparticles allows for retention of nanoparticles with the protein corona during each wash step. This results in a highly specific and reproducible protein corona. Protein coronas were reduced, alkylated, and digested with Trypsin/Lys-C to generate tryptic peptides for LC-MS analysis. All steps were performed in a one-pot reaction directly on the nanoparticles. The in-solution digestion mixture was then desalted, and all detergents were removed using a solid phase extraction and positive pressure (MPE) system on the SP100 Automation Instrument. Clean peptides were eluted in a high-organic buffer into a deep-well collection plate. Equal volumes of the peptide elution were dried down in a SpeedVac (3 h–overnight), and the resulting dried peptides were stored at −80 °C. Using the results from the peptide quantitation assay, peptides were thawed and reconstituted to a final concentration of 50 ng/μL in the Proteograph Assay Kit Reconstitution Buffer. 4 μL of the reconstituted peptides were loaded on an Acclaim PepMap 100 C18 (0.3 mm ID x 5 mm) trap column and then separated on a 50 cm μPAC analytical column (PharmaFluidics, Belgium) at a flow rate of 1 μL/minute using a gradient of 5–25% solvent B (0.1% FA, 100 % ACN) in solvent A (0.1% FA, 100% water) over 22 min, resulting in a 33 min total run time. The peptides generated from these multi-nanoparticle-sampled proteins were analyzed using a dia-PASEF method[12] on a timsTOF Pro 2 mass spectrometer *(Bruker Daltonics)*.

## Peptide and protein quantification
All MS files were processed using the DIA-NN 1.8.1 software[15] and a library-free search with match-between-runs (MBR) enabled against the UniProt database (accessed June 2022) and thereof derived *PAV-exclusive* and *PAV-inclusive* libraries, as described above. Peptide and protein intensities were quantified using the DIA-NN in match-between-runs mode with flags: `--mass-acc-ms1 10`, `--mass-acc 10`, `--qvalue 0.1`, `--matrices,`, `--met-excision`, `--cut K*,R*`, `--smart-profiling`, `--relaxed-prot-inf`, `--reannotate`, `--threads 32`, `--predictor`, `--unimod4`, `--use-quant`, `--peak-center`, `--no-ifs-removal`, and `--reanalyse`.

## Statistical analysis
Statistical analysis was performed using R (version 4.2.1) basic libraries (fisher.test and lm) and Rstudio (version 2023.03.0). Significant MS-PAVs were identified through the construction and analysis of a 2 × 2 matrix. This matrix depicted how many individuals had the genetic variant at a given genomic location, and the corresponding variant peptide. We used the Fisher's Exact test to determine if there was a non-random association between these two categorical variables. Those that were statistically significant at a Bonferroni-corrected alpha level (out of 2341 signals) were considered for future analysis. pQTLs were determined by regressing alternate allele count to protein quantification intensities using a linear regression model. We followed previous analysis protocols[2]; briefly, we used inverse-normalized protein intensities as dependent variables and included age, sex, BMI, diabetes status, and the first three genetic principal components as covariates in a linear model with the copy number of the minor allele as dependent variable.

## Variant annotation
The web servers snipa.org[28], phenoscanner.medschl.cam.ac.uk[23] and omicsciences.org[6] were used to identify previously reported pQTLs

and overlapping information from disease GWAS, and expression and metabolomics QTLs. LocusZoom[29] was used to generate regional association plots.

## APOB analysis

We downloaded UK Biobank GWAS summary statistics for LDL cholesterol (code 30780) and Apolipoprotein B (code 30640) for the joint male/female analysis with from 343,621 individuals from https://github.com/Nealelab/UK_Biobank_GWAS, extracted the +/−250 kb region around variant 2:21263900:G:A and visualized the association data using the LocusZoom server (https://my.locuszoom.org).

## Reporting summary

Further information on research design is available in the Nature Portfolio Reporting Summary linked to this article.

## Data availability

The MS-proteomics data are available via ProteomeXchange with identifier PXD042852 (raw and processed data). Consent obtained from the study participants does not allow deposition of genetic information in public databases. Researcher affiliated with a research institution may request access to genetic data on an individual basis from the corresponding author (Karsten Suhre, Weill Cornell Medicine – Qatar, Doha, Qatar). Access is subject to approval by the institutional research board of Weill Cornell Medicine – Qatar. Source data are provided in the Source Data file. Source data are provided with this paper.

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

## Acknowledgements

K.S. wishes to thank Alex Forest-Hay for initiating this project. We are grateful to all participants of QMDiab for providing their time and blood, and to the late Prof. Mohammed M. El-Din Selim for enabling the sample collection at Hamad Medical Corporation, Doha, Qatar. K.S. and F.S. are supported by the Biomedical Research Program at Weill Cornell Medicine in Qatar, a program funded by the Qatar Foundation. K.S. is also supported by the Qatar National Research Fund (QNRF) grant NPRP11C-0115-180010. The statements made herein are solely the responsibility of the authors.

## Author contributions

Study design: K.S.; Conducted Experiments: G.R.V., H.G., Data analysis: K.S., G.R.V, H.G., S.B., F.S.; Provided Materials: A.H., N.S., G.T., H.S., G.R.V., H.G.; Manuscript writing: K.S.; Manuscript editing: K.S., G.R.V., H.G., K.M., M.D., A.S., S.B., F.S. All authors contributed to the interpretation of results and critically reviewed the manuscript.

## Funding

## Competing interests

G.R.V., H.G., M.D., K.M., A.S., and S.B. are employees and/or stockholders of Seer, Inc.; The other authors declare no competing interests.
