## [Peer Review File · Nature Communications]

Nanoparticle Enrichment Mass-Spectrometry Proteomics Identifies Protein-Altering Variants for Precise pQTL MappingReviewer #1 (Remarks to the Author):

In this study, Suhre and colleagues provide one of the first examples of a genetic study of plasma proteins using mass spectrometry methods, with a large number of peptides (~18,000) and proteins (~3,000), albeit in a small number of people (320). They focus on protein-altering variants, which can be a problem with affinity-binding methods of protein quantification, showing that some of those identified in previous studies are likely to be false positives (which is arguably the most important finding of the paper). They also identify some PAV associations not previously reported, highlighting a potential strength of this novel protein assay platform. The novel methodology for quantifying proteins avoiding the impact of PAVs also represents a useful development for this area of science.

General comments:

1) While the issue of protein-altering variants is indeed a relevant one for most studies, which have used affinity-based assays, it wasn't clear why the authors only focused on PAVs in this study, rather than looking at all possible cis-pQTLs (or even full GWAS), which would have made a more complete and interesting paper

2) The methodology to better quantify proteins without influence of PAVs seems like a promising way to improve protein abundance readouts, thereby enhancing ability to detect associations of proteins with their determinants. To prove this, it would be useful to compare associations of proteins quantified with/without the new methodology with non-genetic determinants (e.g. age, sex, BMI), and other non-PAV genetic associations.

Specific comments:

1) In the Introduction, the authors state that "A recent study showed that approximately 50% of putative epitope-modifying variants colocalize with GWAS associations, suggesting that these variants modify protein properties rather than protein abundance⁷." - Why does this suggest that the variants modify protein properties rather than abundance? Couldn't the cause of the GWAS association be altered protein levels?

2) From Figure 1 (which was very helpful for understanding the study design), I understood that there were 2,899 proteins that could be quantified in the QMDiab samples from the 13,577 with a variant peptide. But only 137 unique genes from the 2,899 proteins had an apparent association signal. What conclusions can be drawn from this low fraction – can the authors suggest how much of this might be due to low sample size, the limited search space (i.e. only the PAVs, rather than regulatory variants), lack of true genetic signal, limitations of the protein quantification etc?

3) There is a much higher proportion of cis pQTLs overlapping with the 184 MS-PAVs using the SomaLogic platform compared to the Olink platform. Does this provide important information about the likelihood of epitope-binding issues in aptamers compared to antibodies?

4) Why did the authors use 20% of samples as the threshold for quantification? How sensitive are the findings to alternative thresholds?

5) The F5 example is a great way of illustrating the issue and the solution. However, it also illustrates that there still appears to be a reasonably high false-positive detection rate, despite steps taken to minimise this problem and focus only on MS-PAVs that suffer less from this issue. In the example given for F5, there were 21 out of 169 false positive detections for one of the variant/reference peptides, and 15 out of 21 false positive detections (if I understood Figure 3 correctly) for the other. Can the authors comment on why there is such a high false positive rate, and whether there are ways that this could be improved? For some of these, there appears to be high levels of the erroneous peptide – although genotyping (and imputation) is considered to be the "ground truth", it would be worth confirming via cluster plots / imputation scores that the genotyping is indeed correct for these individuals.

6) There are multiple reasons why previous studies of Olink and Somascan might not have detected some of the MS-pQTLs reported here. This could be because those platforms don't quantify the relevant proteins, because they do assay them but aren't quantifying them well, or because the PAVs present in the QMDiab population are not sufficiently common in the ancestries used in these other studies. It would be helpful to the reader to get a sense of which of these might be most likely.

7) On a related note, it would also be important to make sure that the replication aspect is comprehensive. For example, the deCODE study of Somascan proteomics (Ferkingstad et al) found an association between rs709932 and SERPINA1, which is listed as not being a cis-pQTL in either Somascan or Olink in Table 1.

8) What r^2 thresholds (and ancestry populations) were used to query the disease databases, such as PhenoScanner? Ideally one would test for colocalization to help confirm that the protein QTL and disease/trait association are likely to be driven by the same causal variant, but I appreciate that this is not straightforward when looking at individual PAVs in a different ancestry population, so being clearer about the thresholds used would help to allow the reader to determine the likelihood of the pQTL driving the trait/disease association.

9) The authors state that "Detections by distinct nanoparticles can be considered technical replicates under different protein extraction protocols and offer additional internal validation of the data." Given that this is the first genetic study of this novel platform, it would be useful to the community if the authors could provide some assessment of how well the replicates across nanoparticles perform.

Reviewer #2 (Remarks to the Author):

Suhre et al studied potential relationships between genetic variants and peptide levels (pQTL) for a large number of proteins in QMDiabetes study. The work is important as most pQTL studies have been based on affinity proteomic methods such as Olink PEA or Somascan aptamers (or single-plex antibody methods). The manuscript is well-written with succinct and economic writing, if still likely somewhat technical for a non-MS scientist. The manuscript also lays out an analytical workflow for peptide-based pQTL discovery, which is more complex and different from workflows for affinity methods.

I have some questions and comments.

1. Row 125: A 10% MAF filter was implemented to ensure there is some detectability of all three possible genotypes. It is stated that "we expect at least 2-3 individuals to be homozygous for the minor allele at this level" Considering that the samples have been genotyped, wouldn't this number be known already so that the minor allele count can be given for all MS-PAV, or calculated as an average MAC?

2. Row 156 / or Discussion: Do I understand correctly that the theoretical presence of a certain peptide was firstly evaluated by the genotype data, and that the presence of the specific peptides was then evaluated in measured peptide data? From a library of 18,000+ peptides covering nearly 3,000 proteins, it seems as if the MS-PAVs identified were 184 for 137 gene products. Even though the sample size in QMDiabetes isn't huge, the proportion of pQTL/proteins tested seems quite small compared to Olink or somascan based pQTL studies. Do the authors think that the lower number is that epitope effects explain many of the Olink and somascan pQTL, or do some pQTL present on affinity platforms remain undetected with the MS-PAV approach?

a. A follow-on question is if the authors can estimate the overlap of proteins tested for pQTL in Pietzner and Sun et al. vs. those tested in the present study i.e. Proteograph?

b. Were there proteins with clear cis-pQTL in affinity platforms that were measured but remained undetected in the present study, despite large effect sizes?

3. Row 242: It would be interesting to understand if there is a difference in the proportions that overlap between Somascan- and Olink-derive pQTL, and the MS-PAV variants.

4. Row 241-248: can these numbers and overlaps be presented in a table format?

5. Row 241: was the overlap with affinity methods studied for a single most-significant pQTL? It's not clear how this was defined. Also, does it make sense to also use coloc when examining overlaps?

6. Row256: Is it counter-intuitive that such a large proportion of MS-PAV associate with eQTL, considering that the eQTL should reflect transcript abundance? Can the authors speculate on potential mechanisms?

7. Table 2. Is it possible to provide also directional information regarding the MS-PAV that are linked to a GWAS trait?

Reviewer #3 (Remarks to the Author):

Suhre and colleagues integrate systematic plasma profiling by LC-MS with genetic information to identify novel genetic determinants of circulating peptides/proteins. Specifically, they identified 184 PAVs located in 137 genes that are significantly associated with their corresponding variant peptides in MS data. As might be expected, a significant number overlap with cis-pQTLs previously identified by affinity proteomics pQTL studies. Further, 54 MS-PAVs overlap with trans-pQTLs identified in the affinity proteomics studies, which they believe identifies the putatively causal cis-encoded protein and provides experimental evidence for its presence in blood. The remaining 36 MS-PAVs have not been previously reported. A particularly novel findings relate to the "incretin pro-peptide (GIP) that associates with type 2 diabetes and cardiovascular disease." The main strength of the manuscript relates to the new enrichment technology upstream of the MS that purportedly facilitates deeper plasma profiling. The authors have also developed new types of analyses to integrate the MS/peptide and genetic information. However,

1. While the authors are admittedly blazing a new trail, the justification for significance in the discovery phase is not clear. The abstract touts the identification of 18,000 unique peptides from almost 3,000 proteins. If the peptide information is really being used to identify the PAVs shouldn't that number be integrated into the definition? Or can some reasonable data reduction algorithm be employed? This is an important issue.

2. Again, while the novelty of the work is appreciated, there is no independent validation cohort of similar genetic background. This is also an important deficit.

3. While it is appreciated that concordance of findings across two different enrichment "chips" should be noted, this does not represent independent findings. "Downstream" MS or data search algorithms are the same.

4. Small point, but why was citrated plasma used for the analyses? Would the analyses be generalizable to all matrices? I would think so, but want to confirm.

5. The data/discussion related to Factor V quantification alone raises a very important omission from the manuscript: why the focus here is the integration of proteomic and genetic data there are many comparisons to affinity based approaches. For context for the reader (and to understand the veracity of the quantification of protein levels), it would be extremely helpful to look at correlations of the levels of the proteins that overlap between the three platforms in these individuals, as well as the associations of the proteins assessed by each platform with very basic clinical traits – such as age, sex, BMI, eGFR. ..and ideally a few gold standard ELISAs..but only if that data were

available.

6. Minor point but what is the throughput of the assay? That is, how much extra time is needed to get at this extra information?

Point-by-point response to the reviewers' comments

Nanoparticle Enrichment Mass-Spectrometry Proteomics Identifies Protein-Altering Variants for Precise pQTL Mapping

K. Suhre et al.

Nature Communications manuscript NCOMMS-23-17240

Reviewer #1

In this study, Suhre and colleagues provide one of the first examples of a genetic study of plasma proteins using mass spectrometry methods, with a large number of peptides (~18,000) and proteins (~3,000), albeit in a small number of people (320). They focus on protein-altering variants, which can be a problem with affinity-binding methods of protein quantification, showing that some of those identified in previous studies are likely to be false positives (which is arguably the most important finding of the paper). They also identify some PAV associations not previously reported, highlighting a potential strength of this novel protein assay platform. The novel methodology for quantifying proteins avoiding the impact of PAVs also represents a useful development for this area of science.

Response: We thank the reviewers for their time and constructive comments and suggestions and hope to have fully addressed all points raised.

General comments:

1) While the issue of protein-altering variants is indeed a relevant one for most studies, which have used affinity-based assays, it wasn't clear why the authors only focused on PAVs in this study, rather than looking at all possible cis-pQTLs (or even full GWAS), which would have made a more complete and interesting paper

Response: We agree that a cis-pQTL study or even a full GWAS with nanoparticle enrichment mass-spectrometry-based proteomics is certainly be of interest for the future. However, we feel that the present sample size of our study is too small for such an undertaking. Also, as we show in our paper, there are data-analysis challenges that need to be overcome. We address here some of these central technological questions (which have been largely ignored by the MS-based proteomics community) and thereby build a foundation for future GWAS-proteomics integration, in which we intend to participate once more data from other studies become available.

2) The methodology to better quantify proteins without influence of PAVs seems like a promising way to improve protein abundance readouts, thereby enhancing ability to detect associations of proteins with their determinants. To prove this, it would be useful to compare associations of proteins quantified with/without the new methodology with non-genetic determinants (e.g. age, sex, BMI), and other non-PAV genetic associations.

Response: We followed the reviewer's suggestion and calculated associations between proteins quantified with/without the new methodology (i.e., using the reference and the PAV-exclusive libraries) and non-genetic determinants age, sex, diabetes state and BMI . We included 3,657 protein group / nanoparticle combinations in the analysis that were detected in more than 20% of shared samples. We found that most proteins (3,183, 87.0%) correlated strongly between the two methods (Spearman rho > 0.8) while only few (91, 2.5%) changed substantially when using the different libraries (Spearman rho < 0.5) [see new Supplementary Figure 5].

We then computed linear models including age, sex, diabetes state, BMI, and the three first genotype principal components as determinants and inverse-normalized protein levels as outcome (analogous to the method in the pQTL analysis). We found many previously reported associations (such as associations between leptin and sex and CRP and BMI) and also new associations that are biologically plausible (such as LIPL and LIPE with diabetes status [see new Supplementary Figure 6]). However, we did not find evidence that using the PAV-exclusive library strengthens the associations between proteins and these non-genetic determinants.

The objective of our study was to improve the associations between genetic variants and protein levels. The fact that we did not see an improvement with non-genetic determinants like age, sex, diabetes status, and BMI is likely because none of these determinants has genetic determinants strong enough that they can be observed in a study of our size.

Nevertheless, the associations between proteins and these non-genetic determinants are of general interest and can serve as additional validation of the platform. As this point has also been raised by reviewer #3, we have added a section describing this analysis to our manuscript and provide the association data in Supplementary Table 4.

Specific comments:

1) In the Introduction, the authors state that "A recent study showed that approximately 50% of putative epitope-modifying variants colocalize with GWAS associations, suggesting that these variants modify protein properties rather than protein abundance7." - Why does this suggest that the variants modify protein properties rather than abundance? Couldn't the cause of the GWAS association be altered protein levels?

Response: In this paragraph, we introduced concerns that epitope-changing variants can suggest changes in protein abundance when, in reality, they only change the affinity binding. What we intended to emphasize there was that at least 50% of these variants still have a biological effect, even if it is not via a change in protein abundance. We realize that the formulation is not conveying this idea correctly and have reformulated the paragraph to:

*"Epitope-modifying variants can result in false-positive associations between genetic variants and protein levels. Additionally, such variants often have a biological impact on the protein function rather than on protein level. A recent study showed that approximately 50% of putative epitope-modifying variants colocalize with GWAS associations, **suggesting that these variants are not mere measurement artifacts.**"*

2) From Figure 1 (which was very helpful for understanding the study design), I understood that there were 2,899 proteins that could be quantified in the QMDiab samples from the 13,577 with a variant peptide. But only 137 unique genes from the 2,899 proteins had an apparent association signal. What conclusions can be drawn from this low fraction – can the authors suggest how much of this might be due to low sample size, the limited search space (i.e. only the PAVs, rather than regulatory variants), lack of true genetic signal, limitations of the protein quantification etc?

Response: There are three reasons why “only” 137 unique genes from the 2,899 proteins had an apparent association signal:

The first reason is that we only include frequent variants (MAF > 10%). The entire protein library contains 61,075 protein entries, of which only 13,577 (22.2%) had at least one peptide with a coding variant at MAF > 10%.

The second reason is that not all theoretically predicted peptides are observable by MS, in most cases either because they do not ionize well or because of lack of a suitable cleavage site. We detected 2,899 proteins, out of which 492 had at least one detectable variant peptide (17.0%). Thus, while there is a variant expected to be present in 22.2% of the detected proteins, only 17.0% were actually found in peptides that were detected with the bottom-up MS workflow utilized in this study.

The third reason is statistical power. Out of the 492 detected variant containing peptides, only 137 were associated with the corresponding genotype at a Bonferroni level of significance. The ones that were detected but were not significant had in most cases less than 10 detections and thereby did not have sufficient statistical power.

Other factors (such as the limited search space - i.e. the fact that only the PAVs, rather than regulatory variants were used, or limitations of the protein quantification) can be ruled out as we are not studying protein expression, but binary detection/non-detection of protein-altering variants. Also, given the present state of the art in genotyping, which has > 99% accuracy, lack of true genetic signal is unlikely a major issue.

We added a paragraph to reflect these considerations.

3) There is a much higher proportion of *cis* pQTLs overlapping with the 184 MS-PAVs using the SomaLogic platform compared to the Olink platform. Does this provide important information about the likelihood of epitope-binding issues in aptamers compared to antibodies?

Response: The number of overlapping *cis*-pQTLs in the updated comparison is now about the same (87 and 89 for SomaScan and Olink respectively, see new Table 2), while the number of non-detected *cis*-pQTLs where the *cis*-protein is assayed on the platform is higher for SomaScan than for Olink (40 versus 11). We believe this is because aptamers are more likely to be “tightly trained” on specific epitopes, while dual antibodies are more likely to interact with any epitope-changing variant, as they cover a larger surface of the protein. SomaScan *cis*-pQTLs are therefore expected to be fewer but stronger, which agrees broadly with our experience. However, we are hesitant to come to firm conclusions about the likelihood of epitope-binding issues in aptamers compared to antibodies, as many other factors may play a role here, and also because both studies are not equally powered.

4) Why did the authors use 20% of samples as the threshold for quantification? How sensitive are the findings to alternative thresholds?

Response: To ensure stable statistics, an ad-hoc threshold requiring detection in at least 20% (< 80% missingness) of samples was chosen. Other studies sometimes use 25%. Below that threshold, statistical power to detect a significant PAV would be low, and a higher threshold might lead to exclusion of true positives. We added Supplementary Figure 1 to show that the number of detections as a function of missingness is quite flat between 20% and 80% missingness and that the sensitivity to the cutoff should be reasonable in that range.

5) The F5 example is a great way of illustrating the issue and the solution. However, it also illustrates that there still appears to be a reasonably high false-positive detection rate, despite steps taken to minimise this problem and focus only on MS-PAVs that suffer less from this issue. In the example given for F5, there were 21 out of 169 false positive detections for one of the variant/reference peptides, and 15 out of 21 false positive detections (if I understood Figure 3 correctly) for the other. Can the authors comment on why there is such a high false positive rate, and whether there are ways that this could be improved? For some of these, there appears to be high levels of the erroneous peptide – although genotyping (and imputation) is considered to be the “ground truth”, it would be worth confirming via cluster plots / imputation scores that the genotyping is indeed correct for these individuals.

Response: The reviewer is correct. In this case, there were 21 out of 169 wild type homozygotes for which peptides apparently carrying the alternate variant were detected, albeit at a low level.

These are most certainly not genotyping errors, but false identifications by the MS analysis software, as this is a known issue with specific amino acids that have similar or weak signals.

There are some isolated extreme cases, like an E>D substitution in Complement Factor H (CFH peptide SPP[E/D]ISHGVVAHMSDSYQYGEEVTYK; Supplementary Figure 7) where, in almost all samples, both genotypes are detected on the peptide level.

Please note that these false positives are not a feature of our method, but a general issue when processing MS signals. They only become prominent here because we separate the mass spectra by genotype.

Indeed, these false positives actually suggest a way to improve peptide identifications in the future by using genotype information at the time of processing the peptide identification in an integrated fashion, as we outline in the discussion.

6) There are multiple reasons why previous studies of Olink and Somascan might not have detected some of the MS-pQTLs reported here. This could be because those platforms don't quantify the relevant proteins, because they do assay them but aren't quantifying them well, or because the PAVs present in the QMDiab population are not sufficiently common in the ancestries used in these other studies. It would be helpful to the reader to get a sense of which of these might be most likely.

Response: To answer this and also several other points raised by the other reviewers, we entirely overhauled the comparison with the Olink and SomaScan platforms. We now use the full summary statistics from the two largest studies with these two platforms, deCODE and UKB-PPP (now with almost 3,000 proteins, published on 4 Oct 2023; in the previous version, we only had access to 1,500 proteins).

In the updated Supplementary Table 3, we now indicate for each PAV variant whether the corresponding *cis*-protein was available in the respective study, and if so, whether the association was significant. Based on this data, we compiled a new Table 2 and summarized the relevant counts in the paper.

7) On a related note, it would also be important to make sure that the replication aspect is comprehensive. For example, the deCODE study of Somascan proteomics (Feringstad et al) found an association between rs709932 and SERPINA1, which is listed as not being a *cis*-pQTL in either Somascan or Olink in Table 1.

Response: Please see above. We believe that by switching to the deCODE and UKB-PPP studies as base for replication, we are now comprehensive. The association between rs709932 and SERPINA1 is now listed as a *cis*-pQTL on both platforms.

8) What r^2 thresholds (and ancestry populations) were used to query the disease databases, such as PhenoScanner? Ideally one would test for colocalization to help confirm that the protein QTL and disease/trait association are likely to be driven by the same causal variant, but I appreciate that this is not straightforward when looking at individual PAVs in a different ancestry population, so being clearer about the thresholds used would help to allow the reader to determine the likelihood of the pQTL driving the trait/disease association.

Response: We now specify in the paper that we used PhenoScanner with an r^2 of 0.8 using LD from the EUR population. However, most of the overlapping pQTLs are now identified on identical SNPs, as we now use full GWAS summary statistics. Therefore, r^2 thresholds should not be an issue. While a more formal colocalization with GWAS hits would be ideal, this represents a large effort, as it requires summary statistics from all of these GWAS. Colocalization also has its caveats, such as dealing with sites that harbor multiple genetic signals and working with associations between different ethnicities.

9) The authors state that “Detections by distinct nanoparticles can be considered technical replicates under different protein extraction protocols and offer additional internal validation of the data.” Given that this is the first genetic study of this novel platform, it would be useful to the community if the authors could provide some assessment of how well the replicates across nanoparticles perform.

Response: We performed the analysis requested by the reviewer. As shown in the new Supplementary Figure 5, the median Spearman correlation between a peptide measured in two or more nanoparticles was $\rho = 0.67$. Data used in Supplementary Figure 5 is in Supplementary Table 4. We report these results in the revised paper.

Reviewer #2

Suhre et al studied potential relationships between genetic variants and peptide levels (pQTL) for a large number of proteins in QMDiabetes study. The work is important as most pQTL studies have been based on affinity proteomic methods such as Olink PEA or Somascan aptamers (or single-plex antibody methods). The manuscript is well-written with succinct and economic writing, if still likely somewhat technical for a non-MS scientist. The manuscript also lays out an analytical workflow for peptide-based pQTL discovery, which is more complex and different from workflows for affinity methods.

Response: We thank the reviewers for their time and constructive comments and suggestions and hope to have fully addressed all points raised.

I have some questions and comments.

1. Row 125: A 10% MAF filter was implemented to ensure there is some detectability of all three possible genotypes. It is stated that “we expect at least 2-3 individuals to be homozygous for the minor allele at this level” Considering that the samples have been genotyped, wouldn’t this number be known already so that the minor allele count can be given for all MS-PAV, or calculated as an average MAC?

Response: The reviewer is correct in stating that the MAC is already known for the individual variants. The purpose of this statement was merely to motivate the choice of a 10% MAF filter cutoff, as there is a tradeoff between comprehensiveness of variants in the search and false discovery rate in the MS. Alternatively, we could have used a MAC filter. As we also filtered variants on pHWE > 1E-6, both approaches should be broadly equivalent.

2. Row 156 / or Discussion: Do I understand correctly that the theoretical presence of a certain peptide was firstly evaluated by the genotype data, and that the presence of the specific peptides was then evaluated in measured peptide data? From a library of 18,000+ peptides covering nearly 3,000 proteins, it seems as if the MS-PAVs identified were 184 for 137 gene products. Even though the sample size in QMdiabetes isn’t huge, the proportion of pQTL/proteins tested seems quite small compared to Olink or somascan based pQTL studies. Do the authors think that the lower number is that epitope effects explain many of the Olink and somascan pQTL, or do some pQTL present on affinity platforms remain undetected with the MS-PAV approach?

Response: Please refer to our response to reviewer #1, who also felt that the number of identified MS-PAVs was low. As we explain there, this number is in line with expectations given the peptide coverage of current MS proteomics and the statistical power of our study.

We now address this point in the discussion as follows:

“The level of PAV detection (137 out of 2,899 quantified proteins) is in line with expectations for several reasons. The entire protein library contains 61,075 protein entries, of which only 13,577 (22.2%) had at least one peptide with a coding variant with MAF > 10%. We detect variant peptides in 492 (17.0%) of the 2,899 proteins. The difference can be explained by peptides that do not ionize well enough to reach the detector or that do not contain suitable cleavage sites with Trypsin/Lys-C enzymatic digestion to form peptides that can be detected with MS. The fact that 137 of the 492 detected peptides reached the required Bonferroni level of significance in the Fisher exact test can be explained by statistical power, as most of the non-significant variant peptides had less than 10 detections.”

a. A follow-on question is if the authors can estimate the overlap of proteins tested for pQTL in Pietzner and Sun et al. vs. those tested in the present study i.e. Proteograph?

Response: We now provide this information in a new Supplementary Figure 4. Please note that also in response to some of the other reviewers' comments, we now use UKB-PPP with the extended Olink platform and deCODE with SomaScan v4 as a reference, not Pietzner et al. anymore. These studies are presently the two most highly powered studies that provide easily accessible summary statistics.

b. Were there proteins with clear *cis*-pQTL in affinity platforms that were measured but remained undetected in the present study, despite large effect sizes?

Response: A substantial fraction of peptides in a protein cannot be measured by MS, such as peptides that do not ionize well enough, or peptides that lack suitable cleavage sites and are too big to be analyzed (e.g., peptides longer than 30 amino acids are generally filtered out in standard protocols). Epitope-changing *cis*-pQTL variants in such peptides may have large effect sizes in affinity proteomics but would not be detected by MS. There are likely many such strong epitope effect-driven *cis*-pQTLs that are found by affinity proteomics and not MS. However, given relatively the low statistical power of QMDiab compared to UKB-PPP, it is hard to say whether a non-detected *cis*-pQTL from affinity proteomics is due to this effect or lack of power.

3. Row 242: It would be interesting to understand if there is a difference in the proportions that overlap between Somascan- and Olink-derive pQTL, and the MS-PAV variants.

Response: This information has been updated and is now in Table 2: The proportion of overlapping PAV proteins is comparable between platforms (127 for 4,660 proteins assayed by SomaScan and 100 for 2,908 proteins assayed by Olink)

4. Row 241-248: can these numbers and overlaps be presented in a table format?

Response: Following the reviewer's suggestion, we added Table 2 to the manuscript.

5. Row 241: was the overlap with affinity methods studied for a single most-significant pQTL? It's not clear how this was defined. Also, does it make sense to also use coloc when examining overlaps?

Response: In the revised version, we determined the overlap using identical variants between studies. While it makes sense to use colocalization in principle when examining overlaps, we feel that this is somewhat of an overkill in our situation. Because of the information we gain from MS, we know that the variant we investigate is the causal one, as the genotype matches the amino acid substitution. For *cis*-pQTLs, we know from experience that a coding variant in the region is almost always the one with the strongest association signal. Colocalization also has its pitfalls, such as sites with multiple genetic signals and ethnicity-specific effects; thus, we think the results would be inconclusive.

6. Row256: Is it counter-intuitive that such a large proportion of MS-PAV associate with eQTL, considering that the eQTL should reflect transcript abundance? Can the authors speculate on potential mechanisms?

Response: We do not think it is counterintuitive that such a large proportion of MS-PAVs are associated with an eQTL. The most likely explanation is feedback of the PAV on the gene expression level. We

previously found that around 50% of the protein-altering variants in affinity proteomics overlap with a GWAS hit and thus have a biological effect, which most likely would be through gene expression. Note that the eQTL lookup through PhenoScanner includes eQTLs from GTeX and therefore covers a broad range of tissues. It may hence also cover feedback that happens in other organs and where the observed blood protein is only a proxy for processes happening elsewhere.

7. Table 2. Is it possible to provide also directional information regarding the MS-PAV that are linked to a GWAS trait?
--

Response: The MS-PAVs are based on a Fisher test and by definition have the same directionality as the variant that links to the GWAS trait. No additional directional information can be derived from it.

Reviewer #3

Suhre and colleagues integrate systematic plasma profiling by LC-MS with genetic information to identify novel genetic determinants of circulating peptides/proteins. Specifically, they identified 184 PAVs located in 137 genes that are significantly associated with their corresponding variant peptides in MS data. As might be expected, a significant number overlap with cis-pQTLs previously identified by affinity proteomics pQTL studies. Further, 54 MS-PAVs overlap with trans-pQTLs identified in the affinity proteomics studies, which they believe identifies the putatively causal cis-encoded protein and provides experimental evidence for its presence in blood. The remaining 36 MS-PAVs have not been previously reported. A particularly novel findings relate to the “incretin pro-peptide (GIP) that associates with type 2 diabetes and cardiovascular disease.” The main strength of the manuscript relates to the new enrichment technology upstream of the MS that purportedly facilitates deeper plasma profiling. The authors have also developed new types of analyses to integrate the MS/peptide and genetic information. However,

Response: We thank the reviewers for their time and constructive comments and suggestions and hope to have fully addressed all points raised.

1. While the authors are admittedly blazing a new trail, the justification for significance in the discovery phase is not clear. The abstract touts the identification of 18,000 unique peptides from almost 3,000 proteins. If the peptide information is really being used to identify the PAVs shouldn't that number be integrated into the definition? Or can some reasonable data reduction algorithm be employed? This is an important issue.

Response: The task of reducing the MS data first to peptides and then to proteins is effectively performed by the algorithms implemented in the MS data analysis software DIA-NN. A PAV is then identified by using a Fisher test to compare the presence-absence of a specific alternate or reference peptide in the DIA-NN output to the presence or absence of the corresponding gene variant in the genotype data. The number of peptides is integrated into the multiple testing correction, as we apply a conservative Bonferroni threshold to call the PAVs. DIA-NN software also uses an FDR method to exclude spurious identifications.

2. Again, while the novelty of the work is appreciated, there is no independent validation cohort of similar genetic background. This is also an important deficit.

Response: Lack of independent validation is an issue in GWAS studies. This is a reason why we did not venture further into the field of GWAS, as suggested by reviewer #1. While independent replication would certainly be nice, we feel that our conclusions are less prone to false positive identifications than GWAS hits, as the significant PAVs constitute an agreement between two fundamentally independent measurements (i.e. DNA sequencing and protein mass spectrometry). Furthermore, we consider the overlap of matching cis-pQTLs from two affinity proteomics studies, deCODE and UKB-PPP, as providing additional confirmation.

3. While it is appreciated that concordance of findings across two different enrichment “chips” should be noted, this does not represent independent findings. “Downstream” MS or data search algorithms are the same.

Response: We agree that these do not represent independent findings, but rather represent technical replicates under different extraction conditions. We believe that we state this correctly by writing:

“Detections by distinct nanoparticles can be considered technical replicates under different protein extraction protocols and offer additional internal validation of the data.”

4. Small point, but why was citrated plasma used for the analyses?

Response: Only citrated plasma samples were available from QMDiab; other matrices (EDTA and Heparin) had been consumed in previous analyses.

Would the analyses be generalizable to all matrices? I would think so, but want to confirm.

Response: While some plasma-type dependent variation can be expected in what concerns the proteins that can be detected and their quantification, our approach itself is not matrix-specific.

5. The data/discussion related to Factor V quantification alone raises a very important omission from the manuscript: why the focus here is the integration of proteomic and genetic data there are many comparisons to affinity based approaches. For context for the reader (and to understand the veracity of the quantification of protein levels), it would be extremely helpful to look at correlations of the levels of the proteins that overlap between the three platforms in these individuals, as well as the associations of the proteins assessed by each platform with very basic clinical traits – such as age, sex, BMI, eGFR. ..and ideally a few gold standard ELISAs..but only if that data were available.

Response: The focus here is the integration of proteomic and genetic data because of our specific interest in the field. We agree that there are many other comparisons to affinity-based approaches that can be performed. Note that the central goal of our paper is not to compare MS and affinity proteomics platforms per se, but specifically to investigate the role of genetic variants in protein quantification using the different methods, with the primary aim of improving the analysis of MS proteomics data and a secondary aim of using the method in larger scale GWAS.

Nevertheless, as suggested by the reviewer, we now provide associations between age, sex, diabetes status and BMI and protein quantifications. As also outlined in our response to reviewer #1, we found many previously reported associations, such as associations between leptin and sex and CRP and BMI, and also new associations that are biologically plausible, such as LIPL and LIPE and diabetes status [see new Supplementary Figure 6]. While we appreciate the reviewer’s interest in comparing non-genetic associations between platforms, we feel that further discussing non-genetic associations from other SomaScan and Olink studies goes beyond the scope of our present paper. As we share our non-genetic association data in Supplementary Table 4, an interested reader can easily integrate and interpret our data together with similar results from other studies and further investigate these questions.

6. Minor point but what is the throughput of the assay? That is, how much extra time is needed to get at this extra information?

Response: The throughput of the assay is defined by the gradient time of the MS experiment. Generally, 15- to 20-minute gradients are used. As the Seer Proteograph technology uses five different nanoparticle extracts, the extra time (and data space) needed has to be multiplied by five. The current study generated 4TB of data and used about 60 days of effective MS machine runtime. The compute time

needed for analysis essentially remains the same (about one hour per sample), which in our case had to be multiplied by three, as we used three different libraries.

Reviewer #1 (Remarks to the Author):

The authors have adequately addressed all my comments, and those of the other reviewers (which were largely similar), to my satisfaction.

Reviewer #2 (Remarks to the Author):

I have now read through responses and the updated manuscript. I think the authors have adequately addressed the points raised.

Reviewer #3 (Remarks to the Author):

I have read through the responses to my prior questions -- as well as the constructive queries from the other reviewers. I am very impressed by the thoughtful responses and don't have anything else to add at this point. This will be a well received paper and a very useful tool for population biology as the throughput improves.